# ATM: Functions of ATM Kinase and Its Relevance to Hereditary Tumors

**DOI:** 10.3390/ijms23010523

**Published:** 2022-01-04

**Authors:** Sayaka Ueno, Tamotsu Sudo, Akira Hirasawa

**Affiliations:** 1Section of Translational Research, Hyogo Cancer Center, 13-70 Kita-Oji-cho, Akashi-shi 673-8558, Japan; tsudo@hyogo-cc.jp; 2Department of Clinical Genomic Medicine, Graduate School of Medicine, Dentistry and Pharmaceutical Sciences, Okayama University, 2-5-1 Shikata-cho, Kita-ku, Okayama 700-8558, Japan; hir-aki45@umin.org

**Keywords:** hereditary tumors, ATM, DNA damage, redox homeostasis, tumor profiling, precision therapy

## Abstract

Ataxia–telangiectasia mutated (ATM) functions as a key initiator and coordinator of DNA damage and cellular stress responses. ATM signaling pathways contain many downstream targets that regulate multiple important cellular processes, including DNA damage repair, apoptosis, cell cycle arrest, oxidative sensing, and proliferation. Over the past few decades, associations between germline *ATM* pathogenic variants and cancer risk have been reported, particularly for breast and pancreatic cancers. In addition, given that ATM plays a critical role in repairing double-strand breaks, inhibiting other DNA repair pathways could be a synthetic lethal approach. Based on this rationale, several DNA damage response inhibitors are currently being tested in ATM-deficient cancers. In this review, we discuss the current knowledge related to the structure of the *ATM* gene, function of ATM kinase, clinical significance of *ATM* germline pathogenic variants in patients with hereditary cancers, and ongoing efforts to target ATM for the benefit of cancer patients.

## 1. Introduction

Ataxia–telangiectasia (A-T) was first reported in 1957 as a familial syndrome characterized by progressive cerebellar ataxia, oculocutaneous telangiectasia, and frequent pulmonary infections [1]. Other abnormalities include radiation sensitivity, premature aging, and a predisposition for developing cancer, mostly of lymphoid origin in those less than 20 years of age and lymphoid tumors and a variety of solid tumors in adults [2]. A-T is a complex disease and not all individuals with this disease have the same clinical presentation, combination of symptoms, and/or laboratory findings. In the classic form, neurological deficits are typically observed in early childhood. In the mild form, the symptoms are usually less severe with later onset, resulting in longer survival. A-T is caused by loss of activity of the ATM protein, and the degree of ATM protein activity is closely associated with the severity of symptoms [3]. A-T is an autosomal recessive disease. Individuals of all races and ethnicities are equally affected, and the prevalence is estimated to be 1–9 per 100,000 people.

A major breakthrough in understating A-T came with the identification of the gene ataxia–telangiectasia mutated (*ATM*) in A-T patients [4]. Mutations in *ATM* occur throughout the gene with no specific areas of susceptibility. Classical A-T is caused by compound heterozygous nonsense mutations or a frame-shift inducing deletions and insertions. Individuals present with milder symptoms if they possess certain missense, in-frame, or leaky splice-site mutations that allow for the production of residual amounts of functioning ATM protein [3].

People with A-T have an increased incidence of cancers of which the majority are lymphoid tumors and breast cancers [5,6]. The cumulative incidence of cancer in A-T patients has been reported to be 38.2% by 40 years of age. Lymphomas and leukemias most often occur in people with classic A-T who are less than 20 years old. In contrast, carcinomas occur on average at 31.4 years of age. Of note, in patients presenting with mild A-T symptoms, the diagnosis of cancer can precede the diagnosis of A-T [7,8]. Carriers are those who have one mutated copy of the *ATM* gene and are generally healthy. However, in 1987, Swift et al. reported that the relative risks of cancer were approximately 2.3 for men and 3.1 for women who were heterozygous for ATM [9]. Recently, multiple studies have documented associations between increased risk of several types of cancers and heterozygous *ATM* germline pathogenic variants (PVs) [10,11,12]. Additionally, numerous recent publications have reported on the varied roles and influences of ATM in multiple cellular processes. In this review, we summarize the molecular biology of ATM, clinical significance of *ATM* germline PVs in patients with hereditary cancers, and new therapeutic strategies for ATM-related cancers.

## 2. ATM Structure

The *ATM* gene was first cloned and reported by Shiloh et al. in 1995 [4]. This gene is located on chromosome 11 (11q22-23) and consists of 66 exons. The *ATM* gene encodes the approximately 350 kDa ATM protein, which consists of 3056 amino acids. ATM protein consists of multiple domains (Figure 1a). ATM belongs to the phosphatidylinositol 3-kinase-related kinase (PIKK) super family that also includes protein kinases such as mechanistic target of rapamycin (mTOR) and DNA-dependent protein kinase catalytic subunit (DNA-PKcs) [4,13]. Similar to other family members, ATM possesses a kinase domain and functions as a serine/threonine kinase. Several hundred ATM substrates have been identified [14,15]. Within the C-terminus, two domains are conserved throughout the PIKK family members: the FRAP-ATM-TRRAP (FAT) domain and FAT C-terminal (FATC) domain. The FAT domain is necessary for ATM dimerization and contains an autophosphorylation site. The FATC domain is stabilized by a disulfide bond and essential for interaction with partner proteins required for ATM activation and control of ATM kinase activity [16]. The large N-terminal region is comprised of repeated units of a helical HEAT repeat motifs and exhibits considerable sequence variation within the PIKK family. These motifs serve as scaffolds for other proteins or DNA, which enables protein–protein or protein–DNA interactions.

During the past several years, the protein structure of ATM has been elucidated (Figure 1b) [17,18]. Under normal cellular conditions, the ATM kinase forms homodimers that are inactive. Recent high resolution cryo-electron microscopic analysis determined the structures of two distinct ATM dimer conformations: a ‘closed’ symmetrical dimer and an ‘open’ asymmetrical dimer (Figure 1c) [17]. The dimer interface consists of upper and lower layers. The upper layer is formed by the interaction between the FLAP region in the kinase domain of one monomer with the FLAP-BE in the other monomer. The lower layer in the dimer interface is formed by the interaction between the M-FAT domains of the two monomers.

In the closed dimer, the active sites of both kinase domains are inaccessible because the PRD in the FLAP of one monomer acts as a pseudo-substrate for the other monomer. As a result, the closed dimeric complex exists as a minimally active enzyme. In contrast, the open dimer does not make full contact between the FLAP region and FLAP-BE, leaving sufficient space for a substrate to bind in the activation site. This structural feature suggests that the enzymatic function of ATM in open dimers are to be of intermediate status.

ATM can be activated under several cellular stress conditions, such as DNA structural changes, oxidative stress, nutrient deprivation, or hypoxia [13,19,20,21,22]. Without the presence of cellular stress, the kinase function of ATM is strictly autoinhibited. Once activated, ATM phosphorylates multiple proteins involved in DNA repair, cell cycle checkpoints, apoptosis, mitochondrial homeostasis, and reactive oxygen species (ROS) regulation. The mechanisms of ATM activation can be classified into two main pathways: the canonical MRE11-RAD50-NBS1 (MRN) complex-dependent pathway, which acts mainly during the DNA damage response, and the non-canonical MRN-independent pathway, which mainly acts during other types of cellular stresses.

## 3. Roles of ATM in DNA Damage Response

It has been estimated that the nuclear DNA in each human cell is subject to as many as 20,000 to 100,000 endogenous or exogenous damaging incidents per day [24,25]. To counteract DNA damage, cells are equipped with repair systems that are specific to different types of DNA lesions. During DNA replication, mis-paired DNA bases are detected and replaced with correct bases by the mismatch repair system [26]. Small chemical alterations of DNA bases are repaired by the base excision repair system, which excises the damaged base [24]. More complex lesions, such as pyrimidine dimers generated by exposure to UV light, are recognized and repaired by the nucleotide excision repair system, which removes approximately 30 base pairs containing the damaged base.

There are two major types of DNA strand breaks: single-strand breaks (SSBs) and double-strand breaks (DSBs). SSBs are the most common type of DNA damage, which occur tens of thousands of times per cell every day and are repaired by the single-strand break repair process [27]. Contrary to SSBs, DSBs occur quite infrequently with a rate typically less than 10 times per cell per day. DSBs are repaired through two distinct pathways: homologous recombination (HR) and non-homologous end joining (NHEJ) [28]. Each DNA repair pathway must be coordinated with a series of signaling responses that arrest cell cycles or trigger cell death in cases of irreparable lesions.

Incorrectly repaired DNA damages become permanent mutations after cell division, and the cell no longer recognizes the damages as errors. Recently, the in vivo mutation rate per cell per division was estimated [29,30,31]. In hematopoietic cells, the mutation rate/division was reported to be 1.2 or 1.14, and in neurons of fetuses 15 to 21 weeks post conception, 1.3 or 1.37.

ATM serves as an important initiator of the DNA damage response. Activated ATM phosphorylates a number of downstream targets that are essential for DSB repair choice and cell cycle inhibition.

### 3.1. MRN-Dependent ATM Activation in Response to DSBs

Having no DNA binding domain to recognize DNA lesions, ATM is recruited to the DSBs through interaction with other proteins (Figure 2). When DSBs occur, MRN complex is recruited to DNA damage sites by γ-H2AX and RAD17, both of which interact with NBS1 [32,33]. NBS1 has an ATM-binding motif at the C-terminus, which is essential not only for efficient recruitment of ATM but is also critical for ATM-mediated signaling [34]. Interaction with both NBS1 and MRE11-RAD50 (MR) is necessary to activate ATM kinase activity [35]. Two arginine residues (Arg2579 and Arg2580) of ATM are essential for interaction with MR.

Upon recruitment by the MRN complex, ATM is activated by several modifications. Ser1981 phosphorylation has been considered as a marker of activated ATM because this residue is immediately phosphorylated after DNA damage [36]. Phosphorylation of Ser1981 is critical to stabilize ATM at the site of DSBs, although it is not essential for ATM kinase function [37]. Autophosphorylation at other sites, including Ser367, Ser1893, Ser2996, and Thr1885, has been reported to be important for ATM kinase function in response to DNA damage [38]. The acetyltransferase TIP60, which is implicated in the DNA damage response, forms a stable complex with ATM through the FATC domain. TIP60 acetylates ATM at Lys3016 [39]. Both the acetylation of Lys3016 and autophosphorylation of ATM play key roles in full kinase activation. These modifications promote monomerization of ATM and facilitates stable binding with substrates.

Activated ATM phosphorylates members of the MRN complex to initiate the downstream signaling pathway [40]. In addition, ATM-mediated phosphorylation of Pellino1 induces Lys63-linked ubiquitination of NBS1, leading to further activation of ATM [41]. RAD17 is an efficient recruiter of the MRN complex and is activated by ATM through phosphorylation of Thr622 [33]. Activated ATM also phosphorylates Ser139 of histone H2AX in the chromatin surrounding DSBs, forming γ-H2AX foci. γ-H2AX recruit mediator of DNA damage checkpoint 1 (MDC-1), resulting in the accumulation of γ-H2AX-MDC1 [42]. MDC1, phosphorylated by ATM, is stabilized on chromatin and NBS1, a member of the MRN complex, which binds to a constitutively phosphorylated site on MDC1. As a result, more MRN complexes are recruited in the vicinity of DSBs. This positive feedback loop between the MRN complex and ATM is essential for efficient signaling.

### 3.2. DSB Repair Choice

There are two major pathways for DSB repairmen: high-fidelity HR and error-prone NHEJ. NHEJ dominates in G1 and the early S phase of the cell cycle, whereas HR reaches peak activity in the mid-S phase [43]. While many regulatory mechanisms influence which of these two pathways is used in DSB repair, ATM plays important supportive roles in repair pathway choice. To enable HR, ATM supports the recruitment of CtIP endonuclease to DSBs and phosphorylates CtIP at Thr859, which then facilitates the recruitment of the nucleases EXO1 and DNA2 and the BLM2 helicase [44]. CtIP is essential for resection of DNA, which is the first step of HR and occurs mainly in the S to G2 phase of the cell cycle. Cyclin-dependent kinases (CDKs) phosphorylate CtIP on Ser327 and Thr847. The Ser327-phosphorylated CtIP recruits BRCA1 and then BRCA1 is phosphorylated and activated by ATM. During the S-phase, ATM also phosphorylates SPOP, a E3 ubiquitin ligase, which causes a conformational change and leads to stabilization of its interaction with TP53 binding protein 1 (53BP1), one of the NHEJ-related scaffolding proteins [45]. The 53BP1-bound SPOP induces polyubiquitination of 53BP1 and elicits 53BP1 extraction from chromatin, which suppresses NHEJ. During the G1 phase, ATM promotes NHEJ through activation of 53BP1. The factors RAP1-interacting factor 1 homologue, shieldin, and Pax-interacting protein 1 are recruited by 53BP1, which together restrict DNA end resection and promotes NHEJ [46]. ATM also phosphorylates DNA-PKcs [47]. DNA-PKcs is recruited to DSBs by the KU70-KU80 complex and in turn recruits other NHEJ proteins, some of which are phosphorylated and activated by ATM [48].

### 3.3. Cell Cycle Inhibition and Apoptosis

To halt cell cycle progression to allow DSB repair, ATM phosphorylates checkpoint kinase 2 (CHK2) on Thr68 and on other residues (Ser19, Ser33/35 or Ser50) in the SQ/TQ cluster domain. Of note, Thr68 phosphorylation is important for full CHK2 activation [49]. Phosphorylated CHK2 monomers dimerize and become active through autophosphorylation of the kinase domain [50]. Once activated, CHK2 phosphorylates many intracellular targets. CHK2 phosphorylates and inactivates CDC25A and CDC25C phosphatases. Active CDC25A and C remove inhibitory phosphates from CDKs that drive cell cycle progression. CDC25A dephosphorylates and activates CDK2, which promotes S-phase entry and progression. Similarly, CDC25C activates CDK1, which promotes G2/M progression. As CDC25C is also regulated by another kinase CHK1, the cell cycle defects observed in ATM-deficient cells are primarily G1/S checkpoint deficiency [51].

ATM also activates a transcription factor p53, which serves as a strong tumor suppressor. ATM phosphorylates and activates p53 on Ser15 and simultaneously phosphorylates its regulatory ubiquitin ligase MDM2 on multiple sites to prevent ubiquitination and proteasomal degradation of p53; this results in p53 activation and stabilization [52]. CHK2 activates p53 through phosphorylation of Ser20 [53]. p53 activation induces G1 cell cycle arrest through transcriptional upregulation of *CDKN1A*, which encodes the cyclin-dependent kinase p21 [54]. Under sustained cell damage, p53 activation induces apoptotic cell death through transcription of pro-apoptotic genes [55].

### 3.4. MRN-Independent ATM Activation in Response to DNA Damage

In response to changes in the chromatin structure caused by hypotonic stress or chloroquine, ATM is activated by the ATM interactor (ATMIN) (Figure 3a) [56]. ATMIN binds to ATM using a C-terminal motif homologous to that of NBS1. ATMIN competes with NBS1, and the absence of either ATMIN or NBS1 increases flux through the alternative pathway [57]. After activation by ATMIN, ATM phosphorylates the downstream proteins, CHK2, p53, and KRAB-associated protein 1(KAP-1) and promotes genomic integrity and cell survival.

SSBs also activate ATM, although the mechanisms are not fully elucidated [19]. ATM-deficient cells are sensitive to hydrogen peroxide and alkylating agents, both of which produce SSBs [16,58]. Unrepaired SSBs induced ATM activation, promoting S-Phase entry delay, which provides additional time for DNA repair before replication [19].

## 4. Roles of ATM in DNA Redox Homeostasis

Oxidative stress can activate ATM independently of DSBs and recruitment of the MRN complex (Figure 3b) [16]. Under an excessive ROS environment, specific enzymes, such as peroxiredoxin 1 or thioredoxin 1 (TRX1), can chemically modify cysteine residues in ATM to form intermolecular disulfide bonds. Several bonds have been mapped in this dimer form, among which the Cys2991 residues of both monomers are essential a mutation at Cys2991 (Cys2991Leu) produced a defect in ATM activation. The formation of disulfide bonds depends on TRX1, which is eventually reduced, leaving ATM in an inactive state [59]. The FATC domain of ATM is critical for activation by oxidative stress, as an Arg3047X mutation in this domain caused defects of ATM activation by ROS [16].

Higher levels of ROS have been widely observed in ATM-deficient cells. These cells show increased sensitivity to oxidative stress compared with ATM-sufficient cells, indicating that the ROS regulation through ATM is essential [60,61,62]. ROS can be generated from oxygen metabolism. Many organelles in cells are involved in the production of ROS, among which mitochondria and peroxisomes are the main endogenous sources of ROS. ATM-mediated responses to oxidative stress operate in both the mitochondrial and cytoplasmic fraction.

### 4.1. Mitochondrial Fraction of ATM

ATM-deficient cells exhibit mitochondrial dysfunction, which includes decreased total mitochondrial DNA levels and mitochondrial mass, lower mitochondrial respiration rates, and less-efficient mitophagy [63,64,65,66,67]. Mitochondria play a crucial role in oxidative phosphorylation and ATP generation, and ATM-deficient cells have significantly reduced ATP levels and survival potential. ATM modulates mitochondrial function through nuclear respiratory factor 1 (NRF1). During exposure to oxidative stress, ATM phosphorylates NRF1, leading to NRF1 homodimerization, nuclear localization, and transcription of genes associated with mitochondrial functions that regulate ROS levels [20]. Mitophagy is a selective process that removes and recycles damaged mitochondria. ATM sequentially activates CHK2 and p53, and activated p53 subsequently induces enhanced translation of denitrosylase S-nitrosoglutathione reductase (GSNOR), resulting in sustained mitophagy [68].

### 4.2. Cytoplasmic Fraction of ATM

In response to elevated ROS, ATM promotes autophagy. Activated ATM phosphorylates liver kinase B1 (LKB1) and subsequently activates AMP-activated protein kinase (AMPK) and tuberous sclerosis complex 2 (TSC2), which induce repression of mTOR complex 1, resulting in the induction of autophagy and a reduction in ROS [69]. In addition, ROS-ATM activated AMPK phosphorylates unc-51-like autophagy activating kinase 1 (ULK1) at Ser317 and Ser777, which also promotes autophagy [70,71]. Glucose depletion and hypoxia also promote ATM-mediated autophagy through a different pathway. Activated ATM phosphorylates CHK2, and activated CHK2 binds and phosphorylates Beclin1 at Ser90/Ser93. Phosphorylation of Beclin 1 suppresses the formation of the Beclin 1–Bcl-2 complex, which normally functions as a negative regulator of autophagy [72].

Peroxisomes generate ROS as a by-product of fatty acid metabolism [73]. To prevent overproduction of ROS, peroxisome clearance is essential. ATM phosphorylates peroxisomal protein PEX5 at Ser141, promotes PEX5 ubiquitination at Lys209, and subsequently triggers autophagy-associated peroxisome degradation, which is a critical process in peroxisome homeostasis [74,75].

### 4.3. Oxidative Stress and DNA Damage

Oxidative stress directly damages guanine residues on DNA, producing the generation 8-hydroxy-2′-deoxyguanosine (8-OHdG) and other DNA adducts [76]. Oxidized DNA bases are removed mainly by base excision repair system, which generates transient SSBs. In A-T cells and ATM-deficient mouse cells, higher levels of oxidative base lesions and SSBs have been reported [77,78]. Treatment with the antioxidant, N-acetyl cysteine, reduced the 8-OHdG levels and the frequency of SSBs in ATM-deficient mouse cells.

## 5. Germline ATM Variants and Hereditary Cancers

Cancer susceptibility is one of the characteristics of A-T and is caused by loss of ATM activity because of homozygous loss-of-function variants in the *ATM* gene. For both A-T patients and carriers of *ATM* heterozygous germline variants, high risks for cancer have been reported [11]. Through analysis of the Exome Aggregation Consortium cohort, excluding The Cancer Genome Atlas (TCGA), *ATM* germline PVs are prevalent in approximately 0.44% of the population, and higher rates are present in certain types of cancer patients [10]. With the widespread advent of next-generation sequencing, a number of genes can be analyzed simultaneously. The use of multi-gene panels (MGPs) in genetic tests has now become mainstream for diagnosing hereditary cancers. PVs in *ATM* are frequently identified through MGP tests with or without PVs in other DNA damage repair-related genes, such as *CHEK2*, *PALB2*, or *BRCA1/2* [79].

Sequencing of DNA from blood samples sometimes incidentally reveals somatic variants in hematopoietic cells that may confound genetic testing results [80,81,82]. Acquired variants have been reported in hematopoietic stem cells. Although the majority of the acquired variants are inconsequential, some variants confer phenotypic advantage to these stem cells, such as increased proliferative potential. As a result, clones with these variants may expand over time, which is known as clonal hematopoiesis (CH) [83]. The *ATM* gene is one of the most commonly altered genes in CH, making it difficult to interpret the results of genetic tests [84]. It is therefore important to have a precise understanding of the diseases associated with germline *ATM* PVs.

### 5.1. Breast Cancer

The frequency of germline *ATM* PVs in breast cancer patients has been estimated to be approximately 1%, and PVs are associated with all subtypes, except triple-negative breast cancer [85,86]. In a meta-analysis of three cohort studies that evaluated the relatives of A-T patients, the estimated relative risk of breast cancer was 2.8 (90% confidence interval (CI), 2.2–3.7; *p* < 0.0001) for *ATM* PV carriers [87]. Other analyses of patients receiving germline MGP genetic testing for cancer predisposition genes demonstrated that *ATM* PVs were associated with a moderate risk of breast cancer with an odds ratio (OR) of 2.03 or 2.78 [11,86].

It has been suggested that there may be substantially higher risk associated with specific *ATM* variants, such as c.7271T>G (p.Val2424Gly). In a multicenter case–control study genotyping 10 rare variants in *PALB2*, *CHEK2*, and *ATM*, strong evidence of an association with breast cancer risk was observed for *ATM* c.7271T>G at an OR of 11.0 (95%CI, 1.42–85.7; *p* = 0.0012) [88]. An analysis of 27 families with *ATM* PVs showed an association between the c.7271T>G variant and increased risk for breast cancer (hazard ratio (HR), 8.0; 95% CI, 2.3–27.4; *p* < 0.001) [89]. In another analysis, individuals with the c.7271T>G *ATM* variant had a higher risk for invasive ductal breast cancer (OR, 3.76) compared with all *ATM* PVs including c.7271T>G (OR, 2.03) [11].

For individuals with *ATM* germline PVs, annual mammograms beginning at 40 years of age, with consideration for annual breast magnetic resonance imaging (MRI), are recommended in the National Comprehensive Cancer Network (NCCN) Guidelines for Genetic/Familial High-Risk Assessment: Breast, Ovarian, and Pancreatic, Version 1.2022 (https://www.nccn.org/home, Last accessed on 29 November 2021). There are no data regarding the benefit of risk-reducing mastectomy, but this procedure may be considered based on family history.

### 5.2. Pancreatic Cancer

Pancreatic cancer is thought to have a familial or hereditary component in approximately 10% of cases. Germline PVs commonly found in pancreatic adenocarcinoma include DNA damage repair genes and mismatch repair genes, most generally in *BRCA2* (2–6%) and *CDKN2A* (1.5–2.5%) [90,91]. In apparently sporadic pancreatic adenocarcinoma patients, *ATM* germline PVs were identified in 1.2% of pancreatic adenocarcinoma patients [92]. In patients with a family history, *ATM* germline PVs were identified in 2.6% of pancreatic cancer cases [90]. A case–control analysis identified association between *ATM* germline PVs and increased pancreatic cancer risk (2.3% of cases and 0.37% of controls; OR, 5.71; 95% CI, 4.38–7.33) [91]. Another analysis of individuals receiving germline MGP testing demonstrated that *ATM* PVs were associated with a high risk of pancreatic cancer with an OR of 4.21 (95%CI, 3.24–5.47; *p* < 0.0001) [11]. In a study comparing germline PV status versus family history without a known germline PV, the cumulative incidence of pancreatic cancer was significantly higher among those with germline PV (HR, 2.85; 95% CI, 1.0–8.18, *p* = 0.05) [93].

Potential benefits of pancreatic cancer screening include downstaging and higher rates of resectability, which may improve mortality rates [94,95]. However, longer-term studies will be required to determine the effects of screening on survival. In NCCN guidelines, it is recommended to consider pancreatic cancer screening using annual contrast-enhanced MRI or magnetic resonance cholangiopancreatography (MRCP) and/or endoscopic ultrasound (EUS) beginning at 50 years of age (or 10 years younger than the earliest exocrine pancreatic cancer diagnosis in the family, whichever is earlier).

### 5.3. Prostate Cancer

A retrospective case–case study including 313 patients with lethal prostate cancer (PrCa) and 486 with low-risk localized PrCa demonstrated that *BRCA1/2* and *ATM* PV rates were higher in patients with lethal cancer than localized cancer (6.07% vs. 1.44% *p* = 0.0007) and highest in patients with lethal PrCa with metastatic disease at the time of diagnosis (8.2%) [96]. The median survival time was significantly decreased in *BRCA1/2* and *ATM* PVs carriers compared with non-carriers (5 years vs. 16 years, *p* < 0.001). This association remained statistically significant after adjusting for race, age, prostate-specific antigen (PSA), and Gleason score at the time of diagnosis. With regard to the *ATM* gene, rates of germline PVs were also higher in patients with lethal PrCa, although the difference was not significant because of the small patients’ number (1.92% vs. 0.41%, *p* = 0.06).

Another study including 5560 PrCa patients and 3353 controls of European ancestry showed that the likelihood of carrying *ATM* PVs was greater in PrCa cases than in controls (1.2% vs. 0.24%, OR = 4.4, 95%CI, 2.0–9.5; *p* < 0.0001) and cases diagnosed before the age of 65 years than those diagnosed after 65 years of age (OR = 4.9, 95%CI, 2.2–11.1; *p* < 0.0001) [97]. *ATM* PVs were enriched in lethal PrCa cases at 1.7% (95%CI, 1.1–2.1; *p* < 0.0001). Furthermore, an analysis of individuals receiving germline MGP testing showed that *ATM* PVs were associated with moderate-to-high risk of PrCa with an OR of 2.58 (95%CI, 1.93–3.44; *p* < 0.0001) [11].

### 5.4. Ovarian Cancer

In ovarian cancer patients, *ATM* germline PVs were found in 0.64–0.87% of cases, which was significantly greater than the 0.19% frequency in the controls from the Exome Aggregation Consortium [98,99]. *ATM* germline PVs were estimated to moderately increase ovarian cancer risk, with an OR of 1.977 (95% CI, 1.33–2.94; *p* = 0.001) [100]. There is currently insufficient evidence to recommend risk-reducing salpingo-oophorectomy.

### 5.5. Melanoma

Recently, a large multicenter melanoma cohort investigating *ATM* germline PVs described *ATM* as a moderate-risk melanoma susceptibility gene [101]. In this study, *ATM* loss-of-function variants were observed in approximately 1% of melanoma patients, which was greater than that observed in samples from the Genome Aggregation Database (0.36%). Another analysis of individuals receiving germline MGP testing revealed that *ATM* PVs were associated with moderate risk of melanoma with an OR of 1.46 (95%CI, 1.18–1.81; *p* = 0.0006) [11].

## 6. ATM Variants in Cancer Precision Medicine

### 6.1. ATM Variants in Clinical Sequencing

Aberrations in the *ATM* gene are commonly observed in cancer. According to TCGA pan cancer studies, *ATM* mutations are found in approximately 5.3% of all cancers and are most common in uterine corpus endometrial cancer (19.1%) followed by bladder urothelial cancer (13.4%) and colorectal adenocarcinoma (13.1%). *ATM* deep deletion is the most common in cervical squamous cell carcinoma or melanoma, with a rate of approximately 3% (cBioportal, https://www.cbioportal.org, Last accessed on 29 November 2021). The sequencing of tumor-derived DNA is conducted primarily to identify biomarkers that can be used for diagnosis, prediction of prognosis, and therapeutic implications. Additionally, sequencing may uncover germline variants associated with hereditary cancer risks [102].

The germline conversion rate (the number of PVs of true germline origin × 100/total number of detected PVs) of the *ATM* gene was reported to be approximately 50% [103]. Despite the high germline conversion rate, the European Society of Medical Oncology Precision Medicine Working Group did not recommend germline-focused tumor analysis for *ATM* because no consensus had been reached on management strategies for the risk within families. The *ATM* gene was not included in the American College of Medical Genetics and Genomics SF v3.0, which was designed “to recommend a minimum list of genes that places limited excess burden on patients and clinical laboratories while maximizing the potential to reduce morbidity and mortality” [102]. However, with growing evidence that *ATM* germline PV carriers are at risk of developing cancer, it may be worth considering germline testing to determine whether the variants detected by tumor sequencing are of germline or somatic origin, particularly in patients with breast, pancreatic, prostate, ovarian cancer, or melanoma.

### 6.2. Targeting ATM-Deficient Cancers

Base modification and single-strand breaks (SSBs) can eventually become DSBs during the DNA synthesis phase of the cell cycle. Thus, increasing the DNA lesions, blocking their repair, or causing the stalling and collapse of replication forks all increase DSBs that cannot be repaired in ATM-deficient cells, eventually leading to cell death. Poly (ADP-ribose) polymerases (PARPs) are key enzymes in the cellular response to single-stranded DNA damage. PARP-inhibitors work by binding to activated PARP and trapping them on the damaged DNA, leading to stalling of the replication fork and accumulation of SSBs. In the DNA damage response, ataxia telangiectasia and Rad3-related (ATR) protein also plays crucial roles. ATR is recruited to single-strand DNA, which are produced at sites of DNA damage or stressed replication forks. ATR interacts with its partner protein and subsequently phosphorylates multiple downstream targets, which are essential in repairing SSBs. Given this rationale, PARP inhibitors and ATR inhibitors can be expected to induce synthetic lethality in ATM-deficient cells; thus, these inhibitors have been evaluated in clinical trials.

In prostate cancer, *ATM* PVs may be either germline or somatic in origin, and are present in 5–8% of castration-resistant tumors overall; this is an enrichment of approximately two-fold over the frequency found in localized prostate cancers, suggesting an association between *ATM* PVs and aggressive disease [104,105,106]. Beyond acting as a potential prognostic biomarker for prostate cancer, *ATM* status may also be predictive of response to novel targeted therapies. Initial trials using olaparib, a PARP inhibitor, for treatment of metastatic, castration-resistant prostate cancer showed an impressive response rate among patients with PVs in the HR repair pathway, including patients with *ATM* deficiency [107,108,109]. However, patients with *ATM* PVs showed a moderate treatment response compared with those with BRCA1/2 PVs [108].

The recent emergence of potent ATR inhibitors has renewed interest in determining the *ATM* gene or ATM protein status in prostate cancer and other malignancies [110,111]. Combining PARP and ATR inhibitors synergistically promoted anti-tumor efficacy against ATM-deficient cancer cells in xenograft and PDX mouse models [112]. Ceralasertib, an ATR inhibitor also known as AZD6738, is currently being assessed in no fewer than 25 phase I and II clinical trials, both as a monotherapy or in combination with chemotherapy, PARP inhibitors, or immunotherapy. A phase IIa multicenter, open-label study is currently ongoing to evaluate the efficacy of ceralasertib against advanced solid tumors with *ATM* mutations (clinicaltrials.gov identifier, NCT 04564027). Furthermore, an initial phase I trial has shown favorable responses of advanced solid tumors to treatment with BAY1895344, another ATR inhibitor, particularly those tumors with deleterious *ATM* mutations and/or loss of ATM protein [113]. With the encouraging clinical responses reported thus far, the results of ongoing trials in patients with ATM deficiency are eagerly awaited.

## 7. Conclusions

Eighty years have passed since A-T was first described. Since the cloning of the *ATM* gene in 1995, a number of studies have demonstrated that ATM plays key roles in DNA damage response and suppression of cancer at early stages. Recent studies have revealed the roles of ATM in response to oxidative stress. The impact of oxidative stress on cancer progression is now becoming clearer. In the last few years, the protein structure of ATM has been elucidated and hopefully there will be further insights into its function in cancer. Several clinical trials have indicated that *ATM* PVs may be useful as biomarkers for targeted therapy.

Currently, the *ATM* gene is included in nearly all multigene panels used in genetic testing for hereditary breast and ovarian cancer. In cases that are negative for *BRCA1* and *BRCA2*, *ATM* is among the most frequently altered genes, with predicted PVs identified in up to 7.8% of cases. Aberrations in the *ATM* gene are also commonly observed in cancer, some of which may be of germline origin. *ATM* germline PVs have been found to be associated with several types of cancer, especially breast and pancreatic cancer. However, due to a lack of precise risk estimates, no consensus on the management guidelines for the variant carriers exists. Therefore, prospective studies to evaluate the full spectrum of disease outcomes are urgently needed.

## Figures and Tables

**Figure 1 ijms-23-00523-f001:**
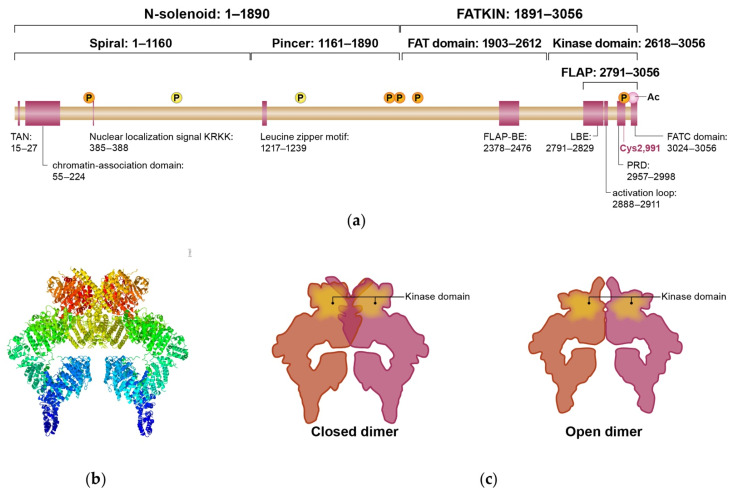
(**a**) Map of the ATM protein, consisting of 3056 amino acids. ATM domains and motifs are listed with their amino acid numbers. Chromatin-association domain serves is important in interacting with chromatin or partner proteins. Nuclear Localization Signal enables nuclear translocation of ATM. Lys3016 is acetylated by TIP60. ATM has multiple phosphorylation sites that can substantially affect its kinase function. Ser367, Ser1893, Thr1885, Ser1981, and Ser2996 are auto-phosphorylated sites. Among them, auto-phosphorylation on Ser367, Ser1893, Ser1981, and acetylation on Lys 3016 are important for ATM activation. FATC domain on C-terminus of ATM is essential for its full activation. Cys2991 is essential to form disulfide bond between two ATM monomers. (**b**) The structure of an ATM closed dimer (PDB ID: 6K9L [23]), created with Jmol, an open-source Java viewer for chemical structures in 3D. (**c**) A schematic representation of the ATM protein. Interface of ATM homodimer consists of upper (FLAP−FLAP-BE) and lower (M-FAT−M-FAT) layers. In the closed dimer, active site of kinase domain is blocked, leaving ATM in an inactive state. In the open dimer, the upper interface is lost, resulting in more compact dimer and allowing partial access to the active site of the kinase domain. FAT-KIN: FAT-phosphatidylinositol 3-kinase-like kinase domain; FLAP-BE: FLAP-binding element; FLAP: Lst8 binding element (LBE), activation loop, and PIKK-regulatory domain (PRD) region.

**Figure 2 ijms-23-00523-f002:**
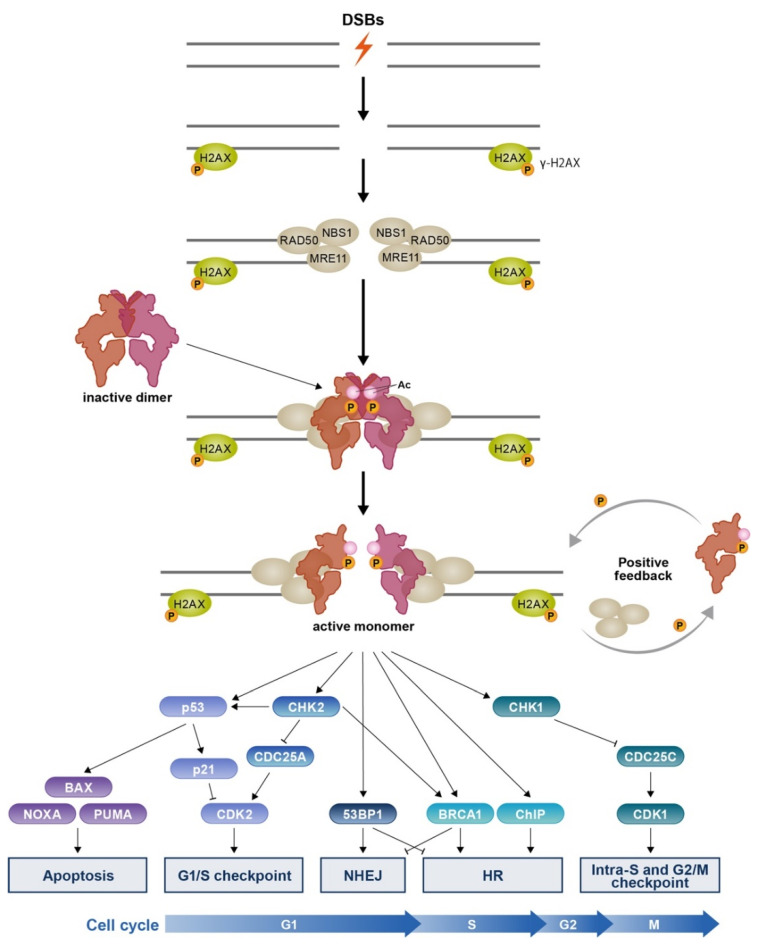
ATM signaling pathway in response to DSBs. ATM is recruited to sites of DSBs by the MRN complex. At the DSB site, ATM undergoes activation through acetylation by TIP60 and autophosphorylation. Direct interaction between the ATM and MRN complex is essential for ATM activation and monomerization. Activated ATM then phosphorylate H2AX surrounding the DSBs, which recruits more of the MRN complex to the site and forms a positive feedback loop between the MRN complex and ATM. ATM phosphorylates and activates a number of downstream targets that are essential for DNA damage repair (NHEJ and HR), cell cycle inhibition, and apoptosis. DNA damages are repaired through either NHEJ or HR in context of the cell cycle state.

**Figure 3 ijms-23-00523-f003:**
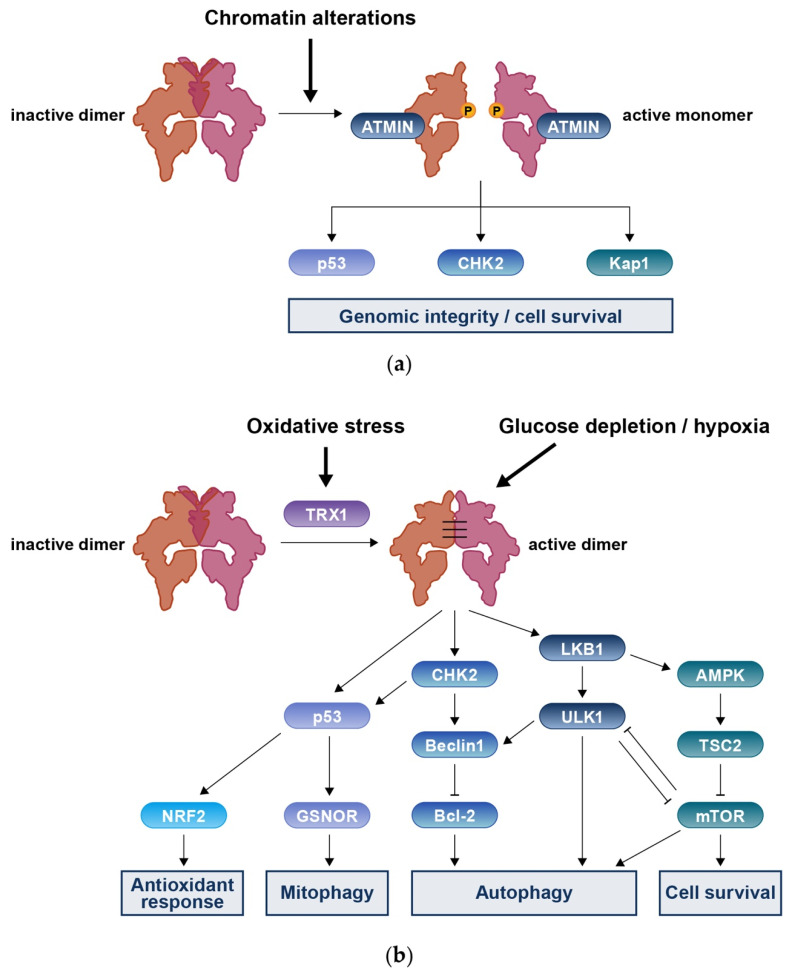
(**a**) Chromatin alterations trigger ATM activation through interaction with ATMIN. ATMIN competes with NBS1 in binding to ATM. Activated ATM phosphorylates and activates p53, CHK2, and Kap1 to promote genomic integrity and cell survival. (**b**) Oxidative stress can also trigger ATM activation through forming intermolecular disulfide bonds in the manner depending on TRX1. Glucose depletion or hypoxia also activate ATM. To reduce oxidative stress or other cellular stress, ATM then activates transcription of genes involved in the antioxidant response. ATM also promotes autophagy and mitophagy to maintain ROS homeostasis while suppressing mTORC1.

## Data Availability

ATM protein structure was obtained from protein data bank (https://www.rcsb.org/structure/6K9L, Last accessed on 29 November 2021).

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
