# Peer review of "ATM: Functions of ATM Kinase and Its Relevance to Hereditary Tumors"

_ijms, 2022, doi:10.3390/ijms23010523_

Round 1

Reviewer 1 Report

In the current review, authors have done a great job comprehensively described the structure of the ATM protein, the functionality of each protein domain, and the role of ATM protein play in DNA repair and damage response and in DNA redox homeostasis. And, more interestingly, authors reviewed the role of germline ATM variants in hereditary cancers, and how ATM has been targeted in precision medicine in treating cancers and how ATM gene is mutated under different treatment strategies. The review was elegantly written, and I believe it will be beneficial to the field of cancer research.

Author Response

Thank you for kind comments. We changed and added sentences to more clearly show the importance of understanding the biology of ATMs in response to the other reviewer's comment. Please see the attachment.

Reviewer 2 Report

The manuscript “ATM: Functions of ATM kinase and its relevance to hereditary tumors” by Ueno et al describes the associations of Ataxia-telangiectasia mutated (ATM) signaling pathways involved in multiple cellular processes such as DNA damage repair, apoptosis, cell cycle arrest, oxidative stress and proliferation that regulate cancer. Overall, the manuscript is well written and authors describe role of ATM in cancer via various pathways. The authors can improve the manuscript by adding some clinical relevance of ATMs and how understanding the biology of ATMs can improve the future perspectives of human diseases such as cancer.

Author Response

Thank you for the constructive comment. We totally agree with your comment. Therefore, we changed and added the sentences in the abstract and in the section, 6.2. Targeting ATM-deficient cancers”. Please see the attachment.
